# Research Progress on the Effect of Nitrogen on Rapeseed between Seed Yield and Oil Content and Its Regulation Mechanism

**DOI:** 10.3390/ijms241914504

**Published:** 2023-09-25

**Authors:** Jiarong Zhu, Wenjun Dai, Biyun Chen, Guangqin Cai, Xiaoming Wu, Guixin Yan

**Affiliations:** The Key Laboratory of Biology and Genetic Improvement of Oil Crops, The Ministry of Agriculture and Rural Affairs of the PRC, Oil Crops Research Institute of the Chinese Academy of Agricultural Sciences, Wuhan 430062, China; 82101215128@caas.cn (J.Z.);

**Keywords:** rapeseed, nitrogen, yield, oil content, molecular mechanisms

## Abstract

Rapeseed (*Brassica napus* L.) is one of the most important oil crops in China. Improving the oil production of rapeseed is an important way to ensure the safety of edible oil in China. Oil production is an important index that reflects the quality of rapeseed and is determined by the oil content and yield. Applying nitrogen is an important way to ensure a strong and stable yield. However, the seed oil content has been shown to be reduced in most rapeseed varieties after nitrogen application. Thus, it is critical to screen elite germplasm resources with stable or improved oil content under high levels of nitrogen, and to investigate the molecular mechanisms of the regulation by nitrogen of oil accumulation. However, few studies on these aspects have been published. In this review, we analyze the effect of nitrogen on the growth and development of rapeseed, including photosynthetic assimilation, substance distribution, and the synthesis of lipids and proteins. In this process, the expression levels of genes related to nitrogen absorption, assimilation, and transport changed after nitrogen application, which enhanced the ability of carbon and nitrogen assimilation and increased biomass, thus leading to a higher yield. After a crop enters the reproductive growth phase, photosynthates in the body are transported to the developing seed for protein and lipid synthesis. However, protein synthesis precedes lipid synthesis, and a large number of photosynthates are consumed during protein synthesis, which weakens lipid synthesis. Moreover, we suggest several research directions, especially for exploring genes involved in lipid and protein accumulation under nitrogen regulation. In this study, we summarize the effects of nitrogen at both the physiological and molecular levels, aiming to reveal the mechanisms of nitrogen regulation in oil accumulation and, thereby, provide a theoretical basis for breeding varieties with a high oil content.

## 1. Introduction

Rapeseed is one of the major oil crops in China. It provides 5.2 million tons of high-quality edible oil annually, accounting for more than 50% of China’s oil crop yield [1]. Although China is a big producer of rapeseed oil and rapeseed meal, its output has been lower than the consumption level for more than ten consecutive years, and so the industry mainly relies on inventory and imports to make up the supply gap. According to statistics, the amount of edible oil imported into the country has been increasing every year, and the country’s self-sufficiency rate in edible oil is seriously insufficient (The National Bureau of Statistics of China. Yearbook of Statistics. http://www.stats.gov.cn/sj/ndsj/ (accessed on 7 September 2023)) (Figure 1). Therefore, the development of rapeseed production is strategically important to maintaining the safety of the national edible oil supply. There is an urgent need to increase the oil production of rapeseed. The oil production depends not only on the seed yield but also on the oil content [2]. Applying nitrogen is an important way of ensuring a strong and stable yield. However, most varieties of rapeseed are extremely sensitive to nitrogen and have a reduced seed oil content after nitrogen application. The oil content has been shown to decrease by 1.6% for every increase of 100 kg·hm^−2^ in nitrogen application [3,4], which equals a decrease of 4% in the seed yield [4,5]. Therefore, to some extent, applying nitrogen fertilizer does not increase rapeseed oil production. It is important to cultivate varieties of rapeseed that do not decrease in oil content after nitrogen application, to ensure oil production.

Nitrogen (N), as a living element of plants, is particularly vital to their growth and development, including photosynthetic assimilation, substance distribution, and the synthesis of lipids and proteins. Plants absorb N from the soil through the plasma membranes of the root epidermis and cortical cells, with nitrates and ammonium being the two main forms of inorganic nitrogen uptake by plants [6]. Nitrogen is absorbed, assimilated, and transported by plants, and it participates in photosynthesis to maintain plant growth [7]. Nitrogen affects photosynthesis by affecting the plant’s ability to synthesize chlorophyll. Furthermore, nitrogen also regulates photosynthesis by affecting the surface area of mesophyll cell walls and chloroplasts [8]. The photosynthetic products sucrose and amino acids are transported from the source leaf to the sink tissue through the phloem and xylem [9]. Sucrose and amino acids are transported in the phloem under the influence of nitrogen [10,11]. After the transition from vegetative growth to reproductive growth, nutrients from the source leaves are transported to the developing seeds for protein and lipid synthesis [12]. However, protein synthesis precedes lipid synthesis, and large amounts of photosynthates are consumed during protein synthesis, which weakens lipid synthesis [13]. Moreover, the molecular mechanism of nitrogen-regulated oil accumulation in developing rapeseed is still poorly understood. Nitrogen levels affect the deposition of proteins and starch in maize and the accumulation of protein and oil in rapeseed [14,15,16]. Due to the different target products, the mechanism of nitrogen on rapeseed is different from its mechanism on cereal crops such as maize and rice. It is important to study the effect of nitrogen on rapeseed grain yield and oil content.

## 2. Effects of Nitrogen on Plants

Sufficient nitrogen will ensure large, vibrantly green leaves, an increase in the number of branches, abundant flowers, and an increase in yield [17]. Excessive nitrogen can hinder the growth and development of plant roots. In *Arabidopsis thaliana*, lateral root elongation will be suppressed with an increasing nitrogen concentration, while primary lateral root growth is insensitive to nitrogen [18]. Too much nitrogen can cause the stems and leaves to swell, reduce the resistance to lodging and stress, and delay the maturation time [16,19,20]. When nitrogen is insufficient, the organic matter in the plant will allocate priority to the root system and stimulate the growth of lateral roots [21]. Thus, the plant grows moderately, diminishes in size, gradually loses fresh leaves, becomes smaller, and has a lower chlorophyll content and redder leaves [22] and a reduction in the number of branches, flowers, and grains [9]. The oil content of maize increases with an increase in the nitrogen application rate [17]. In wheat, the effect of nitrogen on the yield and protein yield forms a quadratic curve [23]. In rapeseed, nitrogen can lead to a significant increase in yield, but can also cause a decrease in oil content. The protein, glucosinolate, and erucic acid content all tend to increase with higher nitrogen levels [2,16,19,24]. These results suggest different mechanisms by which nitrogen affects different plants.

## 3. Effects of Nitrogen on Rapeseed Yield

Nitrogen affects the growth and yield formation of rapeseed. It is reasonable to control the nitrogen application rate within 90~225 kg·hm^−2^ [24,25,26,27]. Appropriate levels of nitrogen can boost the photosynthetic capacity, metabolic levels, and morphogenesis, and increase yield [22]. A shortage of nitrogen can limit the photosynthetic ability, deplete the nutrient supply required for pod development, and reduce the count of productive pods on the main stem and the number of seeds per pod, thereby decreasing both the weight per 1000 grains and the yield [16,28]. Excessive nitrogen can induce over-growth during the vegetative stage and reduce nitrogen-use efficiencies and yield during the maturity stage. This is because pre-flowering rapeseed consists primarily of vegetative morphogenesis, such as stems and leaves. After the crop flowers, nutrients are transported to the grain [16]. An overabundance of nitrogenous fertilizers can cause stems and branches to produce an excessive number of ineffective shoots and branches, and lead to fertilizer waste and ecological destruction [9]. The number of siliques per plant and number of seeds per pod do not significantly increase with the nitrogen application rate beyond 180 kg·hm^−2^ [25,27].

### 3.1. Effects of Nitrogen on Photosynthetic Assimilation of Rapeseed

Nitrogen affects the efficiency of photosynthetic energy transport in leaves, and then regulates the photosynthetic product. After nitrogen is assimilated, it is mainly transported to the mesophyll cells at the top of the plant through the xylem in the form of glutamine, aspartic acid, glutamic acid, and asparagine for carbon assimilation. Leaves are the primary organ of photosynthesis [9]. Chloroplasts are the sole site of photosynthetic carbon assimilation and the primary site of nitrogen assimilation in cells [29]. In chloroplasts, chlorophyll a and chlorophyll b have the function of collecting and transmitting sunlight energy [30]. The energy previously converted into chemical energy by sunlight is temporarily stored in ATP and NADPH for CO_2_ assimilation in dark reactions [31]. CO_2_ is assimilated through the Calvin cycle pathway (C3) [32]. Triose phosphate (TP) is the net product of carbon assimilation and is assimilated through the pentose phosphate reduction pathway (RPPP) [9]. Further, TP is a major precursor to other biosynthetic reactions, including starch and sucrose biosynthesis, nitrogen and sulfur metabolism, fatty acid biosynthesis, cell wall biosynthesis, secondary metabolism, and other metabolic pathways [32]. The CO_2_ assimilation capacity is mainly affected by the nitrogen content of the leaves and is generally positively correlated with the nitrogen content [27]. This is because rapeseed leaves’ sunlight energy capture processes, electron transport processes, and enzymatic reactions in the carbon metabolism require large amounts of nitrogen to synthesize ATP, NADPH, and various proteases [33]. 

In studies of nitrogen-deficient conditions, the chlorophyll content of leaves was shown to decrease, the original chlorophyll in the cytoplasm of mesophyll cells disappeared, fresh chlorophyll could not be formed, and the photosynthetic rate decreased [22,34]. Increasing nitrogen levels can promote chlorophyll synthesis and increase the Rubisco content, while the high activity of Rubisco can promote the photosynthetic rate [8,16,23,35]. In addition, nitrogen can also regulate the CO_2_ concentration in chloroplasts by increasing the surface area of mesophyll cell walls and chloroplasts to regulate the photosynthetic rates [36].

### 3.2. Nitrogen Affects the Substance Distribution of Rapeseed

The amino acids produced in the process of nitrogen assimilation are transported between different organs through the phloem and xylem [37,38]. Phloem mainly transfers amino acids from photosynthetic tissues (source organs) to non-photosynthetic tissues (sink organs) [39]. The phloem sap of rapeseed contains up to 650 millimolars of free amino acids [40,41]. In the source leaves, sucrose and amino acids are excreted from mature and senescent leaves, loaded into the phloem through cytoplasmic pathways or extracellular phloem loading, and transported over long distances to developing sink organs [42]. The transport and distribution of carbon and nitrogen assimilation products, in different parts and at different times, are regulated by the physiological state, source strength, and expression of the transporters. In the vegetative growth phase, the assimilates in the rapeseed leaves are mainly transported to the immature leaves and developing roots [43,44]. During the reproductive growth phase, the assimilates in the leaves are transported to the flowers, siliques, and grains via phloem loaders and transporters, including sugar and amino acid transporters [38]. 

Nitrogen deficiency will accelerate leaf senescence, and result in a decrease in the protein content in leaves. This may change the amino acid transport activity but increase nitrogen remobilization [41,45,46]. Leaf proteins are degraded when carbon and nitrogen are needed to support the growth of the carbon sink organ. The appropriate application of nitrogen can significantly promote the accumulation of carbon and nitrogen products in leaves, pods, and stems, and promote their transfer from source to sink in the late growth stage [16,24,26,47]. The seed is the primary sink tissue for photosynthetic products. Carbohydrates and amino acids are metabolized and then stored in the seed as proteins and oils [48]. To enable the transport and accumulation of carbon and nitrogen products in the seed, the seed coat and seed sink tissue need to coordinate through a membrane that transports sucrose and amino acids [29]. This membrane plays a crucial role in the movement of nutrients between the two tissues, and its effectiveness is closely related to the quality and quantity of seed development. After the nutrients in the seed coat are transmitted through the cytoplasm of the seed coat parenchyma, the assimilates are exported to the extracellular cavity and subsequently absorbed by the developing seeds [29]. 

### 3.3. Molecular Mechanism of Nitrogen Affecting Rapeseed Yield

With the rapid development of molecular biology techniques, and the intensive research on crop nitrogen use efficiency, the genes that regulate plant nitrogen uptake, transport, and utilization efficiency have been discovered (Table 1). External nitrate and ammonium are absorbed by roots and are then assimilated by the nitrate transporter family (NRTs) and the ammonium transporter family (AMTs) in the roots [49,50,51]. Typically, the uptake of NO_3_^−^ by roots is limited. Most NO_3_^−^ is reduced to NH_4_^+^ by nitrate reductase (NR) and nitrite reductase (NIR) in shoots [52]. NH_4_^+^ is then further assimilated into organic forms through the glutamine synthetase/glutamate synthetase (GS/GOGAT) cycle and participates in the carbon metabolism and biosynthesis of other compounds in plants [53,54]. Once NH_4_^+^ is accumulated, plants will produce toxic symptoms (Table 2). A high concentration of NH_4_^+^ as the only nitrogen source will cause stress, and the nutrition provided by NO_3_^−^ or NO_3_^−^ plus a slight amount of NH_4_^+^ is suitable for rapeseed growth [55,56]. In addition, nitrogen levels have been shown to have a significant effect on nitrate reductase (NR) activity in rapeseed, maize, and wheat [16,17,57,58]. Sucrose and amino acids, the products of plant carbon and nitrogen assimilation, are released from mature and aged leaves, packed into the phloem, and transported to the sink tissues by the sucrose transporter family (SUTs/SUCs/SWEETs) and amino acid transporters [48].

Nitrogen levels affect the expression levels of *ZmSWEETs* and *ZmSUTs* in different parts of the maize during the vegetative stage [10]. The expression levels of *ZmSWEETs* and *ZmSUTs* in the leaves were found to be higher under low-nitrogen conditions than in sufficient-nitrogen conditions, while the expression levels of *ZmSWEETs* and *ZmSUTs* in the internodes were higher in sufficient-nitrogen conditions than in low-nitrogen conditions [10]. These results show that sucrose transport from source to sink is more efficient at sufficient nitrogen levels than at low nitrogen levels [10]. Sucrose synthase (SS) is a regulatory enzyme involved in sucrose degradation. It facilitates the decomposition of sucrose transported from the source organ and provides the basic substances for the synthesis and accumulation of fats and proteins in grains [59]. The activity of sucrose synthetase (SS) in rapeseed seeds is related to the nitrogen level, and the SS activity is low under low-nitrogen conditions [60]. Phytohormones are a class of naturally occurring small organic molecules that play an important role in regulating plant growth and development [61]. Nitrogen signaling and metabolic networks are regulated by hormone signals in plants [62,63,64,65]. The gene *Dehydration-Responsive-Element-Binding Protein 1C* (*OsDREB1C*) belongs to the APETALA2/ethylene-responsive element-binding factor (AP2/ERF) family. Its overexpression can significantly increase nitrogen uptake, transport, and utilization efficiency in rice, as well as the number of grains per panicle, grain weight, yield, and harvest index [65].

NO_3_^−^ and NH_4_^+^ can act as signaling molecules that interact with various plant hormone signals to regulate the plant nitrogen uptake, transport, and assimilation. The regulatory factor DULL NITROGEN RESPONSE1 (DNR1) modulates nitrogen metabolism in rice by affecting the auxin biosynthesis and signal transduction pathways [66]. The transcription factor GROWTH-REGULATING FACTOR 4 (OsGRF4) and the growth inhibitor DELLA confer the co-regulation of the absorption, assimilation, and transport of NH_4_^+^, regulate the photosynthetic carbon fixation capacity of crops, and enable a high nitrogen utilization efficiency, increasing the yield of rice and maintaining homeostasis [67]. There have been numerous general studies into the molecular mechanisms behind the effect of nitrogen on production, but most of them have focused on Arabidopsis, maize, rice, and other plants, and extremely little has been carried out in the field of rapeseed.

**Table 1 ijms-24-14504-t001:** The research progress regarding the carbon and nitrogen metabolism genes related to yield.

Genes	Plants	Functions
*LBD* gene family (*LBD37*, *LBD38*, *LBD39*)	Arabidopsis	LBD transcription factor inhibition of anthocyanin biosynthesis and the nitrogen availability signal [68]
*NRT1.1*; *NRT2.1*; *NRT2*	Arabidopsis	Nitrate transport [69,70,71,72]
*MYB*	Soybean; sugarcane; foxtail millet	Regulation of nitrate transporters [73,74,75]
*NLP7*	Arabidopsis	Regulation of nitrate assimilation [76]
		Nitrate sensors and transcriptional activators initiate nitrate-mediated transcriptome signaling [77]
*NLP5*	Maize	Regulation of nitrate assimilation [78]
*CLC* gene family	Arabidopsis	Regulation of nitrate transport [79,80]
*CBL*, *CIPK8*	Arabidopsis	Regulation of nitrate transport and assimilation [81]
*AMT1.3*; *AMT2.1*; *AMT2.2*; *AMT2.3*; *AMT3.1*; *AMT3.2*;	Rice	Regulation of ammonium salt absorption and transport [82]
*SWEETs*	Maize; arabidopsis	Sucrose transport; the expression is affected by nitrogen [10,83,84]
*SUTs*	Arabidopsis; maize	Sucrose transport; the expression is affected by nitrogen [10,85]
*NAS1/NAP1*	Soybean	Regulating the flow distribution of PEP to maintain the carbon and nitrogen balance [86]
*PBF1*	Maize	Regulate carbon and nitrogen metabolism in a nitrogen-dependent manner [14]
*ANR1*	Arabidopsis; chrysanthemum	ANR1-mediated auxin response [18,87]
*DNR1*	Rice	DNR1-mediated auxin response regulates nitrate uptake and assimilation [66]
*GRF4*	Rice	Regulation of nitrogen absorption, assimilation, and transport [67]
*DREB1C*	Rice	Regulation of nitrogen utilization; ethylene-mediated [65]

**Table 2 ijms-24-14504-t002:** Advances in the study of the genes involved in the plant response to nitrogen toxicity.

Genes	Plants	Functions
*STOP1*	Arabidopsis	Negatively regulates AMT1.1 and AMT1.2 [88]
*CAP1*	Arabidopsis	Regulates NH_4_^+^ transport in root hair [89]
*VTC1*	Arabidopsis	Suppressor for NH_4_^+^ efflux, confers plants with NH_4_^+^ tolerance [90]
*OsEIL1*	Rice	OsEIL1 constrains NH_4_^+^ efflux via the activation of OsVTC1-3 [91]
*MYB* family (*MYB28*, *MYB29*)	Arabidopsis	Regulate Fe translocation from root to shoot under NH_4_^+^ conditions to maintain Fe homeostasis [92]
*GSA-1/ARG1*	Arabidopsis	Regulate auxin signaling upon high-NH_4_^+^ conditions [93]
*OsSE5*	Rice	Regulate plant tolerance to NH_4_^+^ [94]

## 4. Effects of Nitrogen on the Oil Content of Rapeseed 

Oil is the ultimate target product of oil crops. Protein is the basic material of the protoplasm. Nitrogen is a major component of protein in plants, and many intermediates of sugar metabolism are major sources of protein synthesis. Studies into oil crops such as soybeans and sunflowers have found that the trade-off between oil and protein may be the regulatory mechanism for their negative response to high nitrogen levels [95,96]. Zhao et al. found that the nitrogen application rate is positively correlated with the protein content, and negatively correlated with the oil content [97]. Comparing different levels of nitrogen, it has been found that the oil content of rapeseed is the highest without nitrogen application, and the oil content gradually decreases, and the protein content gradually increases, with increased nitrogen application [2,60,98]. A low level of nitrogen is adverse to the biosynthesis of protein and glucosinolate, but is beneficial to the formation and accumulation of lipids in seeds [64].

### 4.1. Nitrogen Affects the Synthesis of Lipids and Proteins in Rapeseed

The reason for the negative correlation between oil content and protein content in plants is that the lipids and protein in plants come from the product of glycolysis—pyruvate (PRY)—so there is substrate competition in the synthesis of the two products [99,100]. After the plant enters the stage of reproductive growth, sucrose in the vegetative body is transported to the developing seed for protein and lipid synthesis [101]. Oil is a higher fatty acid glyceride synthesized by fatty acids and glycerol and is stored in seeds in the form of triacylglycerol [102]. The synthesis of lipids in plants is complex and is divided into three main stages. (1) The formation of fatty acids and fatty acid precursor acetoacetyl coenzyme A (acetyl-CoA): sucrose is converted to hexose via glycolysis, and hexose is oxidized to acetyl-CoA and one molecule of CO_2_ [20]. In this process, every two carbon atoms are converted to acetyl-CoA, and one carbon atom (one third of carbon) is lost as CO_2_ [29]. Acetyl-CoA carboxylates to malonyl coenzyme A (malonyl-CoA), which further synthesizes saturated fatty acids in the plastids, or desaturates to form unsaturated fatty acids [103]. (2) The production of 3-phosphoglycerol: in the cytoplasm, hydroxyacetone phosphate produced via the glycolytic pathway is reduced to 3-phosphoglycerol via the catalysis of 3-phosphoglycerol dehydrogenase (G3PDH) [20]. (3) The assembly of fatty acyl-CoA and 3-phosphoglycerol: fatty acyl-CoA and 3-phosphoglycerol were dehydrated and condensed in the endoplasmic reticulum to produce triacylglycerol (TG) [9,104]. In rapeseed, there are numerous enzymes and proteins involved in the synthesis of lipids, and nitrogen is an important element in the formation of proteins. An adequate supply of nitrogen increases protein, and protein synthesis precedes fat synthesis, but in protein synthesis, large amounts of photosynthetic products are consumed, resulting in a weakening of the library source allocated for lipids and, thus, affecting the oil content.

### 4.2. Molecular Mechanism of Nitrogen Level Affecting Lipid Synthesis in Rapeseed

The oil content is a complex quantitative trait, influenced by multiple genes and subject to complex genetic interactions with the environment. The balance between lipid and protein synthesis in plants depends on the activity of key enzymes in their metabolism [101]. Pyruvate carboxylase (PEPC) is a key enzyme controlling the ratio of protein/oil content and is involved in the regulation of seed storage protein and fatty acid metabolism in plant seeds [59,99]. The activity of PEPC may be coordinated through post-translational regulation, with the overexpression of *PEPC* genes accelerated after nitrogen application [15,105]. Acetyl-CoA carboxylase (ACCase) is the rate-limiting enzyme in fatty acid biosynthesis [106]. The expression of the *ACCase* gene in rapeseed treated with different nitrogen changes significantly after 31 days of pollination [15]. After nitrogen application, the expression of the *ACCase* gene decreases in the high-oil-content material, while it increases in the high-protein-content material [15]. Phospholipid phosphatase (PPase) is the rate-limiting enzyme in the biosynthesis of triacylglycerol [59]. The oil content of seeds is related to the activity of the *PPase* gene. The *PPase* activity in oilseeds is highest at 35 and 55 days after flowering under high-nitrogen treatment, second-highest under medium-nitrogen treatment, and lowest under low-nitrogen treatment [60]. Diacylglycerol acyltransferase (DGAT2) catalyzes the synthesis of triacylglycerol (TAG) from diacylglycerol (DAG) in plants and is the last rate-limiting step in TAG synthesis [15]. The overexpression of the *DGAT2* gene in tobacco can increase the oil content and the content of oleic acid and linoleic acid in leaves [15,103]. Similarly, the expression of the *DGAT2* gene in the accession with a high oil content was decreased 12-fold after 31 days of pollination, compared to the case without nitrogen application [15]. Pyruvate kinase (PK) catalyzes phosphoenolpyruvate (PEP) to pyruvate (PYR) and is the last irreversible enzyme in the glycolysis pathway [104]. Glycerin-3-phosphate dehydrogenase (G3PDH) is involved in energy metabolic pathways such as glycolysis, gluconeogenesis, and the Calvin cycle and plays an important role in promoting lipid synthesis [105]. The expression of the *G3PDH* and *PK* genes was found to be enhanced with increasing nitrogen application levels [106]. Nitrogen limitation affects the expression of many genes involved in nitrogen and carbon metabolism during maize grain development, with significant effects on the starch and protein contents [14]. PBF1 promotes the accumulation of zein in the endosperm under sufficient nitrogen conditions and, in the deficiency of nitrogen, PBF1 directs the carbon skeleton used for zein synthesis more toward carbohydrate formation [14]. Although the mechanism of transcription factors regulating oil and protein accumulation under N treatment has not been found in the current studies on rapeseed or other oil crops, the regulatory network behind it is complex and huge, as the oil content in rapeseed is a complex trait. Therefore, the molecular mechanism of nitrogen levels affecting the oil content of rapeseed still needs to be further investigated.

## 5. Discussion and Expectation

Due to the effect of nitrogen on the rapeseed oil content and protein content, a systematic review on nitrogen uptake, transport, assimilation, and utilization in rapeseed and other crops will help us to understand the effects of nitrogen nutrition on rapeseed during the whole growth period (Figure 2). However, the current research on the effect of nitrogen on oil content has the following problems. Firstly, the varieties studied are few and poorly representative. Studies on the effects of nitrogen application on the yield and oil content of rapeseed tend to select only a few materials for comparison, without large batches of material screening and with a lack of broad representation. Secondly, the blind application of nitrogen fertilizer lacks precision. Studies show that the amount of nitrogen can be adjusted according to needs within the range of 0~337.5 kg·hm^−2^ [15,25,27]. In addition, a nitrogen application rate maintained in the range of 90~180 kg·hm^−2^ can produce a higher oil content in the case of an increased yield, but the oil production of rapeseed remains at high values when the rate is 180 kg·hm^−2^ [27]. However, previous studies have used only two or three varieties for cultivation experiments, and different varieties have different nitrogen requirements, meaning that the final results obtained cannot be applied to all varieties of rapeseed. Lastly, there are few studies on the genes that regulate the effect of nitrogen on oil and protein content. Carbon and nitrogen partitioning in rapeseed growth and development is controlled by multiple genes and is involved in multiple metabolic pathways. Four hundred and forty eight candidate genes related to acyl lipids, and eleven genes related to storage proteins in rapeseed have been identified [104]. Previous studies have found that the genes related to lipid synthesis, such as *ACCase*, *PEPC*, and *DGAT2*, indeed react to nitrogen [15,106]. However, the complex regulatory effects of nitrogen on lipid and protein synthesis remain unclear. 

To summarize, it is necessary to screen germplasms with a high oil yield after nitrogen application, and investigate the mechanisms by which nitrogen affects the yield and oil content, to balance the relationship between the yield and oil content, and to reduce nitrogen application without reducing oil production. In the future, to increase oil production through nitrogen application, the following suggestions have been made:

(1) The excavation of rapeseed for carbon- and nitrogen-efficient germplasm resources. There are abundant rapeseed germplasm resources with a high level of genetic diversity in China. It is necessary to ascertain whether any varieties produce an almost non-decreasing effect in oil production upon the application of nitrogen. We can screen high-oil-content materials at high nitrogen concentrations using representatives of the diverse rapeseed natural population and focus on the phenotypic characterization of these materials throughout the whole growth period. This will provide a theoretical basis for the cultivation of a high-oil-yield germplasm after nitrogen application.

(2) Improving the cultivation level and the proper application of nitrogen. The relationship between the protein and oil content should be balanced. Suitable varieties of rapeseed should be selected first, and then the amount of nitrogen should be applied, to improve the yield and quality. For varieties with an insignificant reduction in oil content with nitrogen addition, more nitrogen can be applied, to obtain a higher oil production. For the varieties with an obvious reduction in oil content with nitrogen application, more nitrogen can be applied, but the maximum oil production cannot be achieved.

(3) Understanding the mechanisms of carbon and nitrogen balance and identifying important regulatory genes, and providing a basis for the molecular breeding of high-oil-producing varieties of rapeseed after nitrogen application. In the future, we can identify the genes that control the oil content, and identify the protein content QTL loci using genome-wide association analysis combined with multi-omics analyses, including transcriptome, metabolome, and epigenome analyses. The function of some key genes should be verified using transgenic methods combined with potting or field experiments. In addition, new high-yielding varieties of rapeseed should be developed through molecular design breeding, which is also an essential direction for future research.

## Figures and Tables

**Figure 1 ijms-24-14504-f001:**
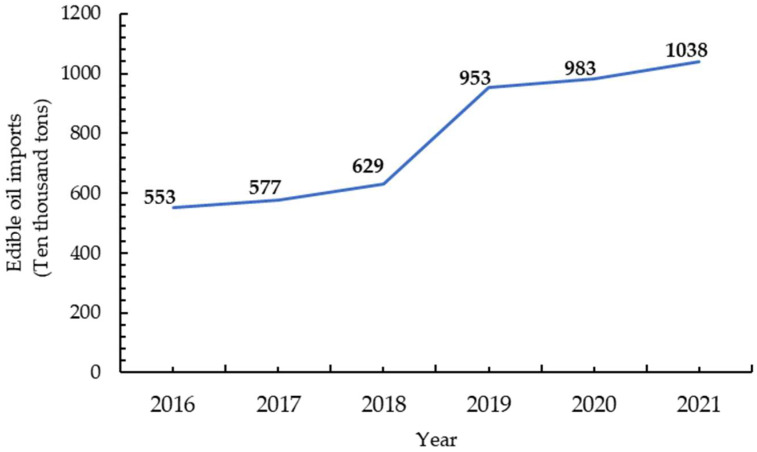
China’s edible oil imports have increased year-on-year since 2016 (data from the National Bureau of Statistics of the People’s Republic of China).

**Figure 2 ijms-24-14504-f002:**
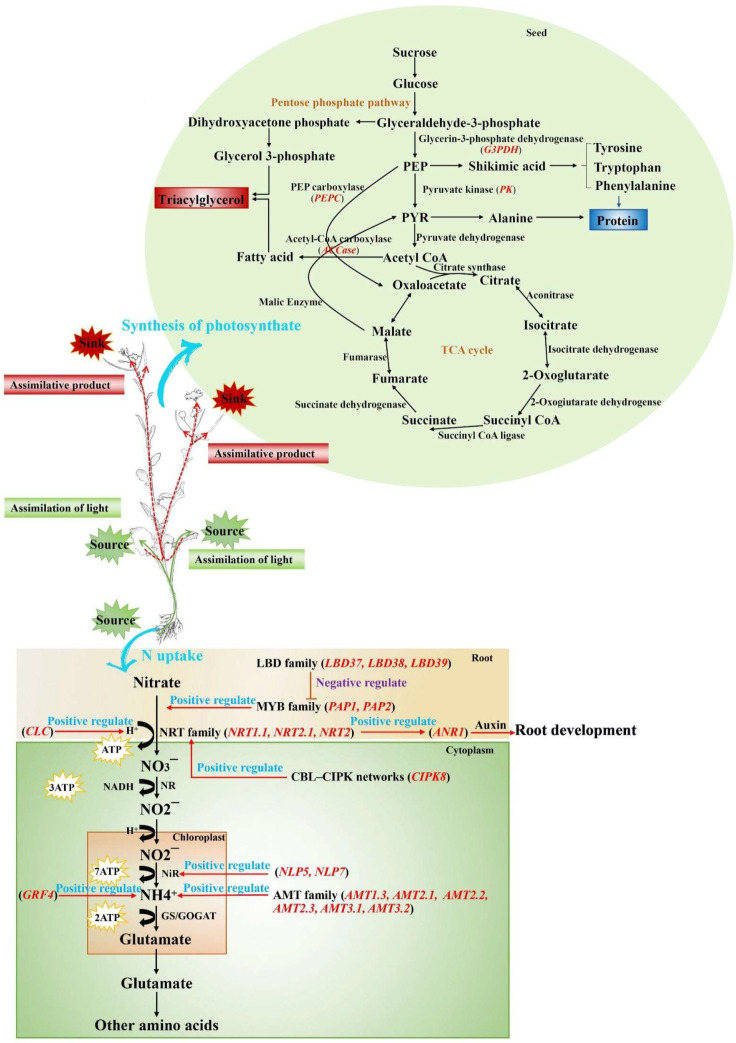
The distribution of carbon and nitrogen source–sink nutrients in rapeseed. The sympathetic transporter (NRT) transports nitrate (NO_3_^−^) to mesophyll cells, and the nitrate reductase (NR) converts nitrate (NO_3_^−^) to nitrite (NO_2_^−^). Nitrite (NO_2_^−^) binding H^+^ is transferred to the stroma of chloroplasts and converted to ammonium (NH_4_^+^) by nitrite reductase (NiR). Ammonium (NH_4_^+^) is converted to glutamate by glutamine synthetase (GS) and glutamate synthetase (GOGAT). Glutamic acid is converted to other amino acids in the cytoplasm [35]. The root absorbs N and transports it to the leaves for photoassimilation, and the assimilated product is transported through the phloem to the developing leaves and growing roots. Nutrients from old leaves are remobilized into developing seeds for lipid and protein synthesis. Red words indicate genes involved in nitrogen regulation, blue words indicate positive regulation, and purple words indicate negative regulation. The green line shows the path of nitrogen transfer, and the red line shows the path of the redistribution of photosynthetic products. PEP: phosphoenolpyruvate; PYR: pyruvate.

## Data Availability

Not applicable.

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
