# Peer review of "Research Progress on the Effect of Nitrogen on Rapeseed between Seed Yield and Oil Content and Its Regulation Mechanism"

_ijms, 2023, doi:10.3390/ijms241914504_

Round 1

Reviewer 1 Report

I have carefully read the manuscript entitled “Research progress on the effect of nitrogen on rapeseed between seed yield and oil content and its regulation mechanism” and analyzed its potential for publication in the MDPI journal International Journal of Molecular Sciences.

In my opinion, the manuscript meets the criteria of reasonable scientific article.

 I suggest accepting the manuscript after minor revision.

 Below you can find my detailed remarks regarding the manuscript.

1. In my opinion, the Introduction is too sparse and does not provide sufficient justification for the importance of the topic undertaken. It would be advisable to include an applicable explanation of this aspect.

2. Fig. 1 contains a large amount of information, some of it is written in small font, which makes the drawing difficult to read. Such an overload of information also does not serve his informative purpose. My suggestion to the authors is to consider splitting this drawing into two separate ones.

Author Response

  1. In my opinion, the Introduction is too sparse and does not provide sufficient justification for the importance of the topic undertaken. It would be advisable to include an applicable explanation of this aspect.

Response: Thank you for your suggestions. I carefully studied the reliable data released by the National Bureau of Statistics, and cited the latest relevant articles to clarify the significance of the research on the topic. The following content and figure were added in the introduction:

Introduction:

Although China is a big producer of rapeseed oil and rapeseed meal, its output has been lower than consumption for more than ten consecutive years, and so mainly relies on inventory and imports to make up the supply gap. According to statistics, the amount of edible oil imported into the country has been increasing every year, and the country's selfsufficiency rate in edible oil is seriously insufficient. (The National Bureau of Statistics of China. Yearbook of Statistics. http://www.stats.gov.cn/sj/ndsj/ (accessed on 7 September 2023)). Therefore, the development of rapeseed production is strategically important to maintain the safety of the national edible oil supply. There is an urgent need to increase the oil production of rapeseed.

Nitrogen (N), as a living element of plants, is particularly vital for their growth and development, including photosynthetic assimilation, substance distribution, and the synthesis of lipids and proteins. Plants absorb N from the soil through the plasma membranes of the root epidermis and cortical cells, with nitrates and ammonium being the two main forms of inorganic nitrogen uptake by plants [3]. Nitrogen is absorbed, assimilated, and transported by plants and participates in photosynthesis to maintain plant growth [4]. Nitrogen affects plant photosynthesis by affecting the ability to synthesize chlorophyll. Furthermore, nitrogen also regulates photosynthesis by affecting the surface area of mesophyll cell walls and chloroplasts [5]. The photosynthetic products sucrose and amino acids are transported from the source leaf to the sink tissue through the phloem and xylem [6]. Sucrose and amino acids are transported in the phloem under the influence of nitrogen [7-8]. After the transition from vegetative growth to reproductive growth, nutrients from the source leaves are transported to the developing seeds for protein and lipid synthesis [9]. However, protein synthesis precedes lipid synthesis, and large amounts of photosynthates are consumed during protein synthesis, which weakens lipid synthesis [10].

Figure:

Figure 1. China's edible oil imports have increased year-on-year since 2016. (data from the National Bureau of Statistics of the People's Republic of China)

  1. Fig. 1 contains a large amount of information, some of it is written in small font, which makes the drawing difficult to read. Such an overload of information also does not serve his informative purpose. My suggestion to the authors is to consider splitting this drawing into two separate ones.

Response: Thanks for the nice comment. To make the figure clear, we adjusted the size and sharpness of the text. Figure 1 is a description of the absorption, transport, assimilation and utilization of nitrogen during the whole growth period of rapeseed, so the entire process may be more complete in one picture.

Reviewer 2 Report

Comments

In general, the submitted manuscript is set out to review the nitrogen effects on the rapeseed’s oil and yield component/content. I trust that the paper has the publication potential, however it should be improved in various aspects, as mentioned in the following comments:

Abstract

Abstract is too short and poorly written. Please review this section and add some more details.

Introduction

1-Introduction is too short. Such a short introduction is not recommended for a review paper. I woul suggest you to add more relevant information from the previous studies.

2-“Effects of Nirogen on plants”, in this section some grammar mistake swere found. Please review the manuscript and improve the language of the manuscript.

3-Table 1: DNR1? What is this? Please explain this term in detail.

4- “Effects of nitrogen on the oil content of rapeseed” this paragraph is too short and poorly written. Please be specific and discuss the previous results in detail. Please add more relevant studies/evidences in your review paper.

5- I woud suggest you to make another table describing nitrogen toxicity in different plants including oilseed crops.

6- Some abbreviations were also noticed in the manuscript. I would suggest you to explain these term in your manuscript as well.

Discussion and expectations:

1-Some gramar mistakes were found in this section. Please revise the manuscript and improve the sentence structure and language of your paper.

2- In my opinion, this section need thorough revision. Discuss the key findings and future directions in detail.

Extensive editing of English language required

Author Response

  1. Abstract is too short and poorly written. Please review this section and add some more details.

Response: Thank you for your advice. We have added our aim of studying the effects of nitrogen on yield and oil content of rapeseed in abstract. In addition, we added a summary of recommendations for future research directions. The following was the revised abstract:

Abstract: Rapeseed (Brassica napus L.) is one of the most important oil crops in China. Improving the oil production of rapeseed is an important way to ensure the safety of edible oil in China. Oil production is an important index that reflects the quality of rapeseed and is determined by oil content and yield. Applying nitrogen is an important way to ensure a strong and stable yield. However, the seed oil content has been shown to be reduced in most rapeseed varieties after nitrogen application. Thus, it is critical to screen elite germplasm resources with stable or improved oil content under high levels of nitrogen and to investigate the molecular mechanisms of nitrogen regulation of oil accumulation. However, few studies on these aspects have been published. In the current review, we analyze the effect of nitrogen on the growth and development of rapeseed, including photosynthetic assimilation, substance distribution, and the synthesis of lipids and proteins. In this process, the expression levels of genes related to nitrogen absorption, assimilation, and transport changed after nitrogen application, which enhanced the ability of carbon and nitrogen assimilation and increased biomass, thus leading to a higher yield. After entering the reproductive growth phase, photosynthates in the body are transported to the developing seed for protein and lipid synthesis. However, protein synthesis precedes lipid synthesis, and a large number of photosynthates are consumed during protein synthesis, which weakens lipid synthesis. Moreover, we suggest several research directions, especially for exploring genes involved in lipid and protein accumulation under-nitrogen regulation. In this study, we summarize the effects of nitrogen at both the physiological and molecular levels, aiming to reveal the mechanisms of nitrogen regulation in oil accumulation and thereby provide a theoretical basis for breeding varieties with high oil content.

  1. “Effects of Nitrogen on plants”, in this section some grammar mistake swere found. Please review the manuscript and improve the language of the manuscript.

Response: Thanks for the helpful comment. we have checked carfully and corrected mistakes in grammar in the whole manuscript.

  1. -Table 1: DNR1? What is this? Please explain this term in detail.

Response: Thanks for the nice comment and good suggestion. We explained the detail of DNR1 in the following:

The regulatory factor DULL NITROGEN RESPONSE1 (DNR1) modulates nitrogen metabolism in rice by affecting auxin biosynthesis and signal transduction pathways [67].

genes

plants

functions

DNR1

Rice

DNR1-mediated auxin response regulates nitrate uptake and assimilation [67]

  1. “Effects of nitrogen on the oil content of rapeseed” this paragraph is too short and poorly written. Please be specific and discuss the previous results in detail. Please add more relevant studies/evidences in your review paper.

Response: Thanks for the nice comment. In this paragraph, we have corrected the grammatical errors. The study about the effects of nitrogen on the oil content of rapeseed is very few. The related studies on other oil crops, such as soybean and sunflower, have been added “Studies of oil crops such as soybeans and sunflowers have found that the trade-off between oil and protein may be the regulatory mechanism for their negative response to high nitrogen[96-97]. Zhao et al. found that nitrogen application rate is positively correlated with protein and negatively correlated with oil content [98]”.

  1. I woud suggest you to make another table describing nitrogen toxicity in different plants including oilseed crops.

Response: Thank you for your suggestions. Your suggestion is very helpful to our understanding of plant nitrogen nutrition. We found that the related studies have mainly focused on plant toxicity caused by ammonium excess. We have added the content of nitrogen toxicity mitigation by plants in Table 2.

genes

plants

functions

STOP1

Arabidopsis

Negatively regulates AMT1;1 and AMT1;2 [89]

CAP1

Arabidopsis

Regulates NH4+ transport in root hair [90]

VTC1

Arabidopsis

Suppressor for NH4+ efflux, confers plants with NH4+ tolerance [91]

OsEIL1

Rice

OsEIL1 constrains NH4+ efflux by activation of OsVTC1-3 [92]

MYB familyMYB28, MYB29

Arabidopsis

Regulate Fe translocation from root to shoot under NH4+ conditions to maintain Fe homeostasis [93]

GSA‐1/ARG1

Arabidopsis

Regulate auxin signaling upon high NH4+ conditions [94]

OsSE5

Rice

Regulate plant tolerance to NH4+ [95]

  1. Some abbreviations were also noticed in the manuscript. I would suggest you to explain these term in your manuscript as well.

Response: Thank you for your meticulous review of our manuscript. We have added the full names of these abbreviations by scrutinizing the manuscript: acetoacetyl coenzyme A (acetyl-CoA), malonyl coenzyme A (malonyl-CoA), 3-phosphoglycerol dehydrogenase (G3PDH), triacylglycerol (TG).

  1. Discussion and expectations: Some grammar mistakes were found in this section. Please revise the manuscript and improve the sentence structure and language of your paper.

Response: Thanks for your careful check. We have checked carefully and corrected mistakes in grammar in the whole manuscript. Language of my paper has been revised in MDPI and obtained the English-Editing-Certificate.

  1. Discussion and expectations: In my opinion, this section need thorough revision. Discuss the key findings and future directions in detail.

Response: Thanks. We have revised the contents of this section and pointed out the problems with the present study of the effect of nitrogen on the yield and oil content of rapeseed in three respects. In addition, we present some suggestions from the perspective of current research directions. The revised content as follows:

Discussion and expectation

Due to the effect of nitrogen on rapeseed oil content and protein content, a systematic review on nitrogen uptake, transport, assimilation, and utilization in rapeseed and other crops will help us to understand the effects of nitrogen nutrition on rapeseed during the whole growth period. However, current research on the effect of nitrogen on oil content has the following problems: Firstly, the varieties studied are few and poorly representative. Studies on the effects of nitrogen application on the yield and oil content of rapeseed tend to select only a few materials for comparison, without large batches of material screening and lacking broad representation. Secondly, blind application of nitrogen fertilizer lacks precision. Studies show that the amount of nitrogen can be adjusted according to needs within the range of 0 ~ 337.5 kg·hm-2 [15, 25, 27]. In addition, a nitrogen application rate maintained in the range of 90 ~ 180 kg·hm-2 can produce higher oil content in the case of increased yield, but oil production of rapeseed at high values when the rate is 180 kg·hm-2 [27]. However, previous studies have used only two or three varieties for cultivation experiments, and different varieties have different nitrogen requirements, so the final results obtained cannot be applied to all varieties of rapeseed. Lastly, there are few studies on the genes that regulate the effect of nitrogen on oil and protein content. Carbon and nitrogen partitioning in rapeseed growth and development is controlled by multiple genes and is involved in multiple metabolic pathways. Four hundred and forty eight candidate genes related to acyl lipids and eleven genes related to storage proteins in rapeseed have been identified [105]. Previous studies have found that the genes related to lipid synthesis, such as ACCase, PEPC, and DGAT2, indeed react to nitrogen [15, 111]. However, the complex regulatory effects of nitrogen on lipid and protein synthesis remain unclear.

To summarize, it is necessary to screen germplasm with high oil yield after nitro-gen application, investigate the mechanisms by which nitrogen affects yield and oil content, to balance the relationship between yield and oil content, and to reduce nitrogen application without reducing oil production. In the future, to increase oil production through nitrogen application, the following suggestions have been made:

(1) Excavation of rapeseed for carbon- and nitrogen-efficient germplasm re-sources. There are abundant germplasm resources of rapeseed with much genetic diversity in China. It is necessary to ascertain whether any varieties produce an almost non-decreasing effect in oil production upon application of nitrogen. We can screen high-oil-content materials at high nitrogen concentrations using representatives of the diverse rapeseed natural population and focus on phenotypic characterization of these materials throughout the whole growth period. This will provide a theoretical basis for the cultivation of high-oil-yield germplasm after nitrogen application.

(2) Improving the cultivation level and the proper application of nitrogen. The relationship between protein and oil content should be balanced. Suitable varieties of rapeseed should be selected first, and then the amount of nitrogen should be applied to improve the yield and quality. For varieties with insignificant reduction of oil content with nitrogen addition, more nitrogen can be applied to obtain more oil production. For the varieties with obvious reduction of oil content with nitrogen application, more nitrogen can be, applied but the maximum oil production cannot be obtained.

(3) Understanding the mechanisms of carbon and nitrogen balance and identifying important regulatory genes, and providing a basis for molecular breeding of high-oil-producing varieties of rapeseed after nitrogen application. In the future, we can identify the genes that control oil content, and identify protein content QTL loci using genome-wide association analysis combined with multiomics analyses, including transcriptome, metabolome, and epigenome analyses. The function of some key genes should be verified using transgenic methods combined with potting or field experiments. In addition, new high-yielding varieties of rapeseed should be developed through molecular design breeding, which is also an essential direction for future research.

Round 2

Reviewer 2 Report

Accept!